# Molecular Characterization, Oxidative Stress-Mediated Genotoxicity, and Hemato-Biochemical Changes in Domestic Water Buffaloes Naturally Infected with *Trypanosoma evansi* Under Field Conditions

**DOI:** 10.3390/pathogens14010066

**Published:** 2025-01-13

**Authors:** Waqas Ahmad, Muhammad Yasin Tipu, Muti ur Rehman Khan, Haroon Akbar, Aftab Ahmad Anjum, Muhammad Ovais Omer

**Affiliations:** 1Department of Pathology, Faculty of Veterinary Sciences, University of Veterinary and Animal Sciences, Lahore 54000, Pakistan; waqasahmadvet@gmail.com (W.A.);; 2Livestock and Dairy Development Department, Government of Punjab, Lahore 54000, Pakistan; 3Department of Parasitology, Faculty of Veterinary Sciences, University of Veterinary and Animal Sciences, Lahore 54000, Pakistan; drharoonakbar@uvas.edu.pk; 4Institute of Microbiology, Faculty of Veterinary Sciences, University of Veterinary and Animal Sciences, Lahore 54000, Pakistan; 5Department of Pharmacology and Toxicology, Faculty of Biosciences, University of Veterinary and Animal Sciences, Lahore 54000, Pakistan

**Keywords:** *Trypanosoma evansi*, genotoxicity, ITS-1 primer, oxidative stress, buffalo health, hematobiochemical markers

## Abstract

(1) Background: Surra is a debilitating disease of wild and domestic animals caused by *Trypanosoma evansi* (*T. evansi*), resulting in significant mortality and production losses in the affected animals. This study is the first to assess the genetic relationships of *T. evansi* in naturally affected buffaloes from Multan district, Pakistan, using ITS-1 primers and evaluating the effects of parasitemia and oxidative stress on DNA damage and hematobiochemical changes in infected buffaloes. (2) Methods: Blood samples were collected from 167 buffaloes using a multi-stage cluster sampling strategy, and trypomastigote identification was performed through microscopy and PCR targeting RoTat 1.2 and ITS-1 primers. Molecular characterization involved ITS-1 via neighbor-joining analysis. The impact of parasitemia loads was correlated with oxidative stress markers, genotoxicity, and hematobiochemical parameters using Pearson correlation and multivariable regression models. (3) Results: Field-stained thin blood film microscopy and molecular identification revealed 8.98% and 10.18% infection rates, respectively. Phylogenetic analysis based on ITS-1 region sequences of the identified isolates showed close genetic associations with Indian isolates. The mean trypomastigote count observed in the infected buffaloes was 5.15 × 10^6^ (±5.3 × 10^2^)/µL of blood. The parasitemia loads were significantly correlated with the alterations in oxidative stress markers, DNA damage, and changes in hematobiochemical parameters. Infected animals exhibited significant (*p* < 0.05) alterations in oxidative stress biomarkers, including catalase, nitric oxide, and malondialdehyde concentrations. Noteworthily, a comet assay revealed a significantly (*p* < 0.0001) higher mean genetic damage index in the infected buffaloes (0.7 ± 0.04) compared with the healthy ones (0.196 ± 0.004). Alongside significant (*p* < 0.05) reductions in red cell indices, a marked elevation in leukocyte counts and serum hepatic enzyme levels was recorded in the affected buffaloes. (4) Conclusion: *T. evansi* isolates of buffaloes from Multan, Pakistan, have genetic similarities to Indian isolates. This study also revealed that higher parasitemia loads induce genotoxicity in the infected animals through oxidative stress and cause hematobiochemical alterations under natural field conditions.

## 1. Introduction

The livestock sector is an important part of Pakistan’s economy [1]. Dairy animals including buffaloes, cattle, sheep, goats, and camels are chiefly raised in tropical and subtropical climate conditions of the country [2]. *Trypanosoma evansi* (*T. evansi*) is a hemoflagellate parasite which is responsible for “surra,” which significantly affects both wild and domesticated animals. Mechanical transmission of *T. evansi* occurs through biting flies and is commonly found in horses, cattle, buffaloes, dogs, and deer [3]. The disease poses a significant economic burden to livestock farmers by reducing milk and meat production. While several species of Trypanosoma are known, *T. evansi* is the most prevalent in various regions worldwide as well in Pakistan [4,5]. Buffaloes are the most vulnerable to *T. evansi* infection compared with other livestock species [5].

Although PCR testing is relatively costly compared to conventional blood film microscopy, it is highly sensitive and accurate for the detection of trypanosomal DNA in blood samples, and it is an effective method for diagnosing trypanosomiasis across all stages of the disease [6]. Among the genetic targets, ribosomal DNA, internal transcribed spacer (ITS) regions, kinetoplast DNA, and variable surface glycoprotein (VSG) are reliable for PCR-based diagnosis [7]. Particularly when morphological identification is challenging, the ITS-1 and ITS-2 regions are very useful for species identification and exploration of the phylogenetic relationships. Phylogenetic analysis of ITS sequences has been used to study the polymorphism of *T. rangeli*, *T. brucei*, *T. congolense,* and *T. evansi* [7].

Infected animals show a considerable decline in red blood cell count, hematocrit, and hemoglobin concentrations [5]. Previous research has demonstrated that trypanosomiasis not only affects blood cells but also damages various vital organs [8]. Noteworthily, the parasite also causes oxidative stress in buffaloes [5]. Oxidative stress is known to be one of the key mechanisms behind the pathophysiology of trypanosomiasis [9]. Various researchers have proposed that trypanosomiasis induces oxidative damage in the heart, liver, and blood cells of the hosts. Infection with *T. evansi* causes the generation of various toxic free radicals (reactive oxygen species; ROS), including superoxide radicals, hydrogen peroxide, and hydroxyl radicals [8,10]. The activity of these free radicals results in the dysfunction of essential biomolecules including proteins, lipids, and especially DNA damage, leading to cell death [9,11]. Oxidative stress-induced DNA strand break is a key contributor to genetic mutations in living organisms, and over one hundred different types of DNA adduct (including purines, pyrimidines, and the deoxyribose backbone) resulting from oxidative injury have been identified [10].

The alkaline comet assay is a convenient and accurate method to detect disruptions in DNA integrity in cells [12]. Nitric oxide (NO) is commonly used to assess oxidative stress as it reacts with ROS and biomolecules, including nitrites/nitrates and superoxide anion. Changes in catalase (CAT) activity and malondialdehyde (MDA) levels are frequently observed in both natural and experimental *T. evansi* infections [5]. Research has also shown that experimental infections with *Leishmania chagasi* and *T. cruzi* lead to DNA damage in peripheral blood cells [8,13].

There is a scarcity of data regarding the presence of trypanosomiasis in the Multan district, Pakistan. Moreover, existing data on its genotoxic effects in naturally infected hosts are also very scarce in the literature. To address this gap, the present study is the first to explore and identify the genetic relationship of *T. evansi* isolates using the ITS-1 primer from buffaloes of the Multan district, Pakistan. This study also aimed to evaluate the impact of varying *T. evansi* parasitemia loads and infection-induced oxidative changes on DNA damage in peripheral blood lymphocytes’ hematobiochemical alterations in infected buffaloes under natural field conditions.

## 2. Materials and Methods

### 2.1. Study Area and Sampling Strategy

This study was conducted in the Multan district, located in Southern Punjab, Pakistan, between latitudes 29°19′11″ and 30°28′16″ N and longitudes 70°58′34″ and 71°43′25″ E. The Multan district is one of the oldest urban centers in this region, serves as the district’s capital, and comprises four main tehsils: Multan City, Multan Saddar, Shujaabad, and Jalalpur Pirwala (Figure 1). The terrain of this district is predominantly plane and agriculturally productive. The district is bordered by the Chenab River in the west. Multan experiences an arid climate, with an extremely hot summer and moderate winter. In winter, temperatures can drop to 4.5 °C, and summer temperatures peak at 50 °C [14].

A randomized multi-stage cluster sampling pattern was adopted to collect samples from 167 buffaloes kept in traditional farming systems which are common in the district. The Multan district was intentionally chosen, with the study specifically conducted in Multan Saddar tehsil (an administrative sub-division of the district). This tehsil contains 174 villages, each representing a substantial herd. In the second stage, considering a 99% confidence level, a 50% prevalence rate, an intra-class coefficient of 0.4, and a 10% desired absolute precision, 16 clusters were encompassed in the sampling [15]. Since the exact number of buffalo was unknown at the village level, the sample size (*n* = 167) was then calculated based on an expected 50% prevalence, a 10% desired absolute precision, and a 99% confidence interval. In the third stage, systematic sampling was not practical given the local conditions, and because husbandry practices were consistent across the study population, it was assumed that convenience sampling would not affect the validity of the findings [5]. Therefore, a convenience sample of approximately 10 animals was performed from each cluster.

### 2.2. Sample Collection and Parasitological Identification

Blood from the marginal ear vein of the buffalo was collected as per the prescribed procedures [16]. For trypomastigotes identification, thin blood films of the fresh blood were prepared in duplicate and subjected to field staining following the standard procedure [17]. The field-stained thin (FST) blood films were then examined under the oil immersion lens of a light microscope, and approximately 50–100 fields per slide were observed.

### 2.3. Genotypic Confirmation and Molecular Characterization

Genomic DNA was extracted from anticoagulant-added (EDTA K_3;_ 1.2 mg/mL) whole blood using a commercial kit (Gene JET Genomic DNA Purification Kit #KO721). DNA quality was checked by electrophoresis on an agarose gel. Previously reported primer sets, including ITS-1 and RoTat 1.2, were used, targeting the product sizes of ribosomal RNA and variant surface glycoprotein genes, respectively. The primer sequences and product sizes are given in Table 1.

Briefly, PCR was performed in a total 50 µL of reaction mixture having 25 µL master mix, 2 µL of each forward and reverse primer, 2 µL template DNA, and 19 µL of deionized water. The amplification was performed in a Veriti 96-Well Thermal Cycler (Applied Biosystems™, Waltham, MA, USA) using the cycling condition as previously described [18,19]. Amplified products were examined by agarose gel electrophoresis (1.3%). A 100 bp ladder was used to identify the size of the amplified products (Thermo scientific^®^, Waltham, MA, USA) using the Gel Documentation System (Omega Flour^Plus^, San Francisco, CA, USA). The amplified products were sequenced by 1st BASE DNA Sequencing Services (Seri Kembangan, Malaysia), and the sequences were then proofread using BioEdit software (version 7.7.1). A BLAST search was performed, and sequences from around the globe were aligned with the study sequences with CLUSTAL W in MEGA 11 software. The phylogenetic analysis was inferred using the neighbor-joining method through the 1000 bootstrap method [20].

### 2.4. Hematology and Serum Biochemistry

For hematological indices, the whole blood samples collected in EDTA-coated vacutainers were analyzed using an automatic hematology analyzer (Beckman Coulter, Brea, CA, USA), while the specimens without added EDTA were centrifuged at 3000× *g* for 15 min and stored at −20 °C for biochemical investigation [12], including total serum protein (TSP) [21], albumin [22], alanine aminotransferase (ALT), alkaline phosphatase (ALP) [23], aspartate aminotransferase (AST) [24], and gamma-glutamyl transferase (GGT). The globulin concentration was estimated by subtracting the albumin concentration from the TSP [25]. Serum NO levels were estimated as described by Menaka, et al. [26]. The plasma MDA and CAT levels were estimated through colorimetric procedures of thiobarbituric acid [27] and a commercial kit (CAT100-1KT, Sigma- Aldrich^®^, St. Louis, MI, USA), respectively.

### 2.5. Single Cells Gel Electrophoresis

A comet assay was performed on peripheral blood lymphocytes which were purified from the anticoagulant-added whole blood samples. Briefly, 3 mL of blood and 4 mL of Histopaque media were mixed and centrifuged at 25 °C and 800× *g* for 45 min. The isolated lymphocytes were aspirated, mixed in Roswell Park Memorial Institute 1640 Medium (RPMI), and then centrifuged at 250× *g* for 10 min. The comet assay was conducted using the alkaline method [28]. Electrophoresis was performed for 20 min and 300 mA at 25 Volts (0.83 V/cm) in the dark to prevent DNA damage. The prepared slides were exposed to chilled water and prepared for microscopy, and then cells from the slides were observed. The genetic damage index was estimated using the formula described by Ahmad, et al. [12].$$\mathrm{GDI}=\frac{\left(No.of\;cells\;in\;class\;I\right)+\left(2\times No.of\;cells\;in\;class\;II\right)+\left(3\times No.of\;cells\;in\;class\;III\right)+\left(4\times No.of\;cells\;in\;class\;IV\right)}{No.of\;cells\;in\;class\;0+No.of\;cells\;in\;class\;I+No.of\;cells\;in\;class\;II+No.of\;cells\;in\;class\;III+No.of\;cells\;in\;class\;IV}$$

### 2.6. Statistical Analysis

The results regarding the incidence of infection were described using descriptive statistics. The association of demographic variables with the infection was checked by the Chi-square test. The adjusted odd ratio and 95% confidence interval (95% CI) were calculated using the binary logistic regression model. A comparison of the value of the hematobiochemical parameters and the genetic damage index was performed using an independent *t*-test. The Pearson correlation data regarding the relationship of parasitemia loads, oxidative stress biomarkers, and hematobiochemical parameters were integrated using hierarchical clustering. In order to draw the association dendrograms, Euclidean distances and complete linkage were used as metrics and an amalgamation rule, respectively. A multivariable regression model using a stepwise backward elimination method with k-fold cross-validation was employed to assess the impact of parasitemia load and oxidative stress profile on hematobiochemical changes and the genetic damage index. The data analysis was performed in RStudio (version 2024.9.0.375), and graphs were generated using GraphPad Prism 10.3 (GraphPad Software, LLC). The level of significance was kept at α = 0.05.

## 3. Results

### 3.1. Parasitological Identification

The overall 8.98% incidence of trypanosomiasis based on thin blood microscopy and 10.18% with PCR was recorded in the present study of buffaloes from the Multan district. Out of 116 females and 51 males, 10.34% (12 females) and 9.8% (5 males) were found to be infected. A non-significant variation in infection status was recorded regarding gender and age (Table 2).

All of the buffaloes positive for trypanosomiasis through the ITS-1 set of primers were confirmed for the presence of *T. evansi* using RoTat 1.2 primers (Figure 2).

### 3.2. Molecular Characterization

Three different *T. evansi* PCR products using ITS-1 (480 bp) primers were randomly selected for sequencing. The selected sequences from PCR products were submitted to GenBank (accession numbers: PQ764013, PQ764014 and PQ764015). The sequences were blasted for *T. evansi* in various hosts from other different countries. Phylogenetic trees constructed from these sequences are shown in Figure 3. The bubaline isolates demonstrated a close genetic relationship with Indian isolates. The first sequence was closely related (99% site convergence percentage) to an Indian canine isolate (OQ376669.1). Similarly, isolate 2 aligned closely with a Thai bubaline isolate (MN121260.1), showing a convergence of 95%. Lastly, isolate 3 showed a 99% site convergence with another Indian bovine isolate (MN196603.1).

### 3.3. Correaltion of Parasitemia Loads and Oxidative Stress Profile

The mean trypomastigote count recorded in the infected buffaloes in the present study was 5.15 × 10^6^ (±5.3 × 10^2^)/µL. In comparing oxidative stress markers between healthy and infected groups, serum CAT activity was significantly reduced (*p* < 0.0001) in the infected group (2.9 ± 0.07 U/L) compared to healthy controls (5.79 ± 0.04 U/mg). Conversely, NO (µmol/L) and MDA (nmol/L) levels were significantly (*p* < 0.0001) higher in the infected group compared to the healthy group, indicating increased oxidative stress due to infection. Furthermore, the results of the Pearson correlation revealed that serum CAT had a strong negative correlation with trypomastigote count (r = −0.92), while positive correlations with trypomastigote count were recorded for both NO and MDA levels (r = 0.871 and r = 0.898, respectively) (Figure 4).

### 3.4. Single-Cell Gel Electrophoresis

The results of single-cell gel electrophoresis on peripheral lymphocytes revealed a significantly higher mean genetic damage index in the infected group (0.7 ± 0.04) compared to the healthy group (0.196 ± 0.004, *p* < 0.0001) (Figure 5).

Noteworthily, significant differences in DNA damage were observed between the healthy and infected groups. A marked increase in the fraction of cells with higher DNA damage classes in infected individuals was recorded (Table 3).

### 3.5. Hematology and Serum Biochemistry

Regarding hematological indices, the RBC counts, Hb concentration, PCV (%), and MCHC were significantly (*p* < 0.001) decreased, while MCV was increased significantly (*p* < 0.001) in *T. evansi*-infected buffaloes. Decreased MCHC and increased MCV were suggestive of hypochromic macrocytic anemia in the infected buffaloes. Additionally, leukocyte counts were also significantly (*p* < 0.05) higher in the infected buffaloes compared with the healthy ones (Figure 6).

No significant difference in total serum protein concentrations was observed between the groups. However, the levels of serum albumin were significantly decreased and globulins were significantly higher in infected buffaloes than in healthy buffaloes. A significant decrease in albumin to globulin ratio was also noteworthy in the *T. evansi* infected buffaloes. The serum concentrations of various hepatic enzymes, including alanine aminotransferase, aspartate aminotransferase, alkaline phosphatase, and γ-glutamyl transferase were significantly increased in the infected buffaloes (Figure 7).

### 3.6. Impact of Oxidatice Stress Profile on Hematological Biomarkers

The correlation analysis of trypomastigote loads and oxidative stress markers, including NO and MDA levels, showed a significant negative relationship with red cell indices. Conversely, a positive correlation of these markers was observed with genetic damage and serum levels of hepatic enzymes. Notably, CAT had a negative correlation with genotoxicity and positive correlations with RBC indices, highlighting its role in counteracting oxidative damage (Figure 8).

The findings of the multiple regression analysis showed that the higher trypomastigote counts significantly impacted various hematological and biochemical parameters and resulted in increased genotoxicity. Specifically, elevated trypomastigote levels were associated with decreased hemoglobin (*p* = 0.03). Furthermore, higher oxidative stress levels significantly contributed to immune cell activation and hepatic damage. Notably, the interaction effects between trypomastigote count, NO, MDA, and CAT also had an association with genotoxicity, suggesting oxidative DNA damage in response to infection severity.

## 4. Discussion

Pakistan’s tropical and subtropical climate supports various blood protozoan infections, including *Trypanosoma evansi*, which often affect dairy livestock [29]. Such infectious diseases present serious health challenges and impact animal productivity. In this study, the *T. evansi* prevalence detected via field-stained thin blood film microscopy (8.62%) was significantly lower than that identified through molecular techniques (10.34%). Previous research has similarly reported *T. evansi* infection among buffaloes, equines, and camels raised in various climates across Pakistan [5,21,30,31]. No significant associations between the infection and buffalo age or sex were found. Our results align with those of Hussain, et al. [5] who reported similar trends and prevalence rates in buffaloes in Pakistan’s Lodhran district, which is adjacent to our study area.

Molecular characterization using ribosomal DNA sequences is crucial for the phylogenetic classification of trypanosomes [32]. In the present study, we reported the genetic variability among *T. evansi* isolates from the Multan district, Pakistan, by targeting the ITS-1 region of the ribosomal DNA. Although genetic heterogeneity within Trypanosoma species from different hosts has previously been examined in Pakistan, this study is likely the first to report ITS-1 gene sequences of *T. evansi* isolates from buffaloes in Pakistan [33,34,35]. Analysis of these sequences showed a close genetic relationship to *T. evansi* isolates from India [36]. However, Kumar, et al. [36] suggested that the ITS-1 region may not effectively describe the genetic diversity within clonal populations of *T. evansi* from a single host. Similarly, in our study the NJ-consensus tree showed the least genetic diversity among *T. evansi* isolates. This may be attributed to the limited sequence length and high similarity within this region. Broader insights might be gained by examining the complete ITS region, which includes ITS-1, 5.8S, and ITS-2 [37].

Blood leukocytes are highly vulnerable to oxidative damage due to their membrane’s high content of unsaturated fatty acids [38]. Oxidative stress in infected animals results in the weakening of the defense mechanisms of blood cells against pathogens by impairing cell membrane integrity [39]. Several studies have revealed that infection with certain parasites including *Toxoplasma gondii*, *T. cruzi*, and *Leishmania chagasi* can also lead to DNA damage [40,41,42,43,44,45]. Notably, our study is the first to report a correlation between genotoxicity and *T. evansi* infection in buffaloes under natural field conditions. Our findings revealed significant DNA damage in the blood cells of animals infected with *T. evansi*. DNA is a critical target of oxidative stress [46]. Free radicals interact with cellular DNA, leading to damage that ranges from sugar and base alterations to strand breaks and cross-linking with proteins [47]. Increased DNA damage alongside insufficient repair mechanisms is frequently observed in the pathology of numerous infectious diseases [48]. During *T. evansi* infection, inflammatory responses activate cells that induce several oxidant-generative enzymes (i.e., cholinesterase), which are elevated in response to the infection [49]. This activation is aggravated by pro-inflammatory cytokines like IL-1, TNF-α, IL-4, IL-6, and IFN-γ [50]. Baldissera et al. (2014) reported raised expression of these cytokines in rats infected with *T. evansi*. Elevated cytokine production prompts enzymes that generate free radicals and reactive oxidants, such as NO, ROS, H_2_O_2_, and O2•^-^ [51]. These reactive species, though intended to neutralize the parasite, can in excess damage inflamed tissues and induce cellular DNA damage [52].

Following infection, *T. evansi* initially multiplies and disseminates through the host’s bloodstream [53]. The host’s cellular immunity is critical in combating *T. evansi*, with macrophages releasing NO, tumor necrosis factor, and ROS as primary defenses [53,54]. Although these mechanisms aim to eliminate the trypomastigotes, they concurrently expose the body’s organs to genotoxic mediators [54]. At controlled levels, NO is beneficial in host defense due to its brief activity and low concentration. However, excessive NO can induce oxidative stress and DNA strand breaks through the formation of ROS (including peroxynitrite), leading to the initiation of oxidative changes in nucleic acids [55]. Felizardo, et al. [56] reported the elevation of NO levels in the murine models of trypanosomiasis. This NO elevation is commonly observed in the serum of infected hosts, suggesting its dual role, i.e., aiding immune response at low concentrations but promoting oxidative damage when elevated.

In this study, a significant reduction (*p* < 0.05) in RBC counts, Hb concentration, and MCHC among the infected animals was recorded. Noteworthily, the increase in mean corpuscular volume (MCV) and decrease in MCHC among *T. evansi*-infected buffaloes indicated macrocytic and hypochromic anemia. Anemia consistently occurs in trypanosomiasis across all the host species [57]. In trypanosomiasis, various factors acting individually or collectively contribute to anemia. The first possible pathway could include erythrocyte damage from trypanosome flagellar movement, intermittent fevers, platelet aggregation, toxins, metabolites, lipid peroxidation, and malnutrition resulting in hemolytic anemia [58]. Secondly, oxidative stress-induced lipid peroxidation results in the compromised integrity of erythrocytes and causes their destruction [59]. In this context, MDA is one of the most commonly used markers for assessing lipid peroxidation [60]. In this study, MDA levels were significantly elevated in infected buffaloes compared to healthy ones. The findings are aligned with the previous reports. This elevation in MDA may also contribute to macrocytic hypochromic anemia [5]. Another possible mechanism leading to anemia in trypanosomiasis involves the action of sialidase enzymes produced by *T. evansi*, resulting in extravascular hemolysis in the mononuclear phagocytic system [61].

Leukocyte counts were significantly elevated (*p* < 0.05) in the infected buffaloes. Our findings coincide with previous studies that reported increased counts of white blood cells in trypanosomiasis in infected animals. Additionally, the significantly elevated concentrations of various serum enzymes observed in the infected buffaloes may also be attributed to heightened oxidative stress, which can lead to cellular damage, hypoxic conditions, and centrilobular liver degeneration.

## 5. Conclusions

In conclusion, *T. evansi* isolates from naturally infected buffaloes in the Multan district, Pakistan, showed close genetic relationships with Indian isolates, suggesting a possible regional lineage. Elevated *T. evansi* parasitemia loads correlated strongly with increased genotoxicity in peripheral blood lymphocytes, which is primarily driven by elevated oxidative stress. Additionally, the higher trypomastigote counts were associated with significant hematobiochemical changes such as macrocytic hypochromic anemia, leukocytosis, hyperglobulinemia, and elevated hepatic enzyme levels.

## Figures and Tables

**Figure 1 pathogens-14-00066-f001:**
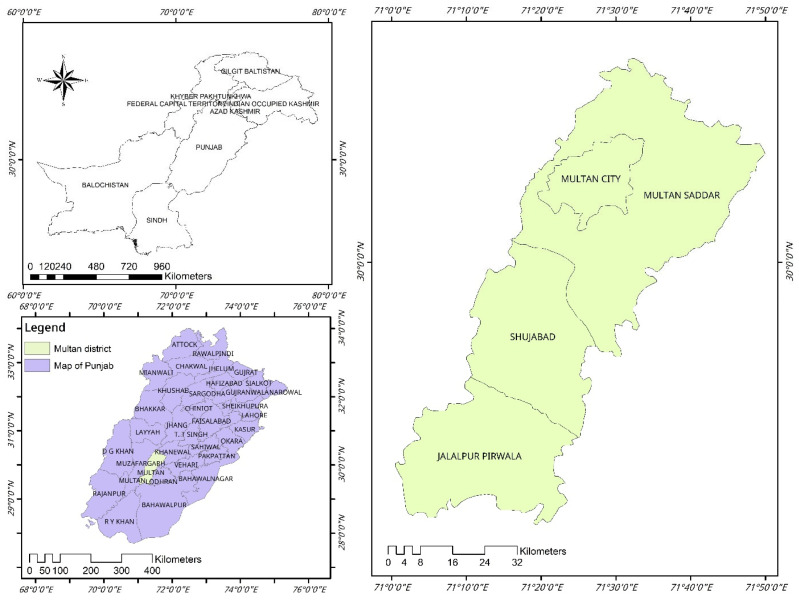
Geographical map highlighting the study area in green with designated sampling sites marked by red circles. The inset map shows location of the Multan district, within Punjab, Pakistan, for regional context. The map was created using ArcMap 10.7.1.

**Figure 2 pathogens-14-00066-f002:**
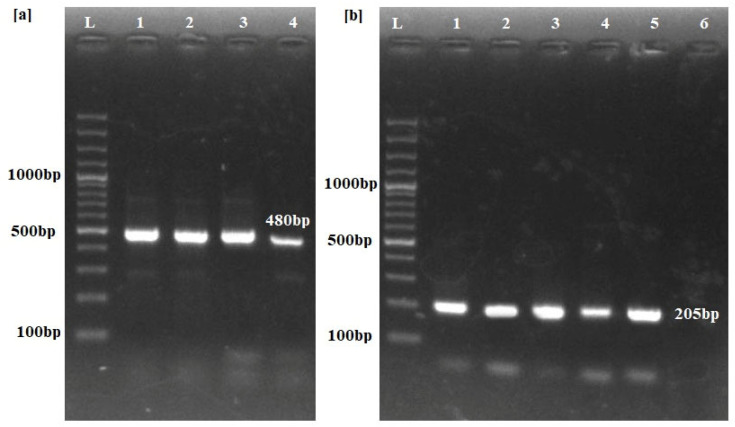
PCR amplification results for identification of Trypanosoma species. (**a**) ITS-1 primers produce a 480 bp band across the positive samples (Lanes 1–4) only. (**b**) Lanes 1–5 show Trypanosoma evansi-type A-specific RoTat 1.2 primers yielding a 205 bp band, confirming T. evansi infection in the positive samples. Lane 6 depicts the results from uninfected buffalo. L: Molecular marker of 100 bp.

**Figure 3 pathogens-14-00066-f003:**
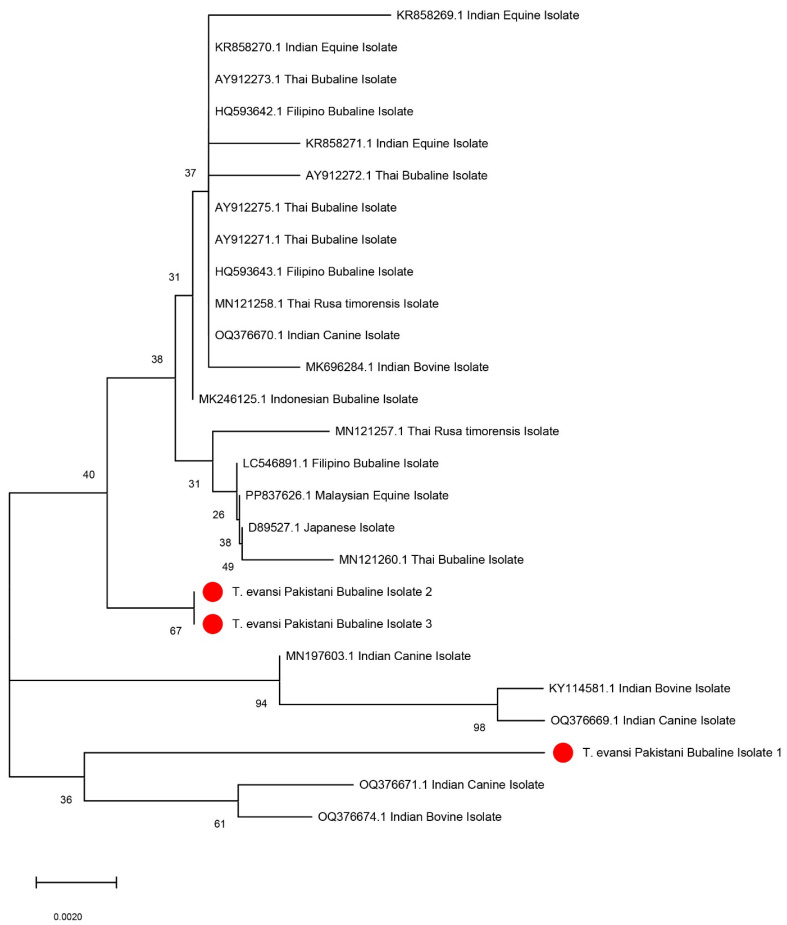
Neighbour-joining phylogenetic tree showing the relationships between *T. evansi* isolates from different geographical and host origins, including the Pakistani isolates sequenced in this study (highlighted in red). The tree is based on partial sequence analysis of the internal transcribed spacer-1 gene, and the bootstrap values (indicated as percentages) at nodes represent support from 1000 replicates. The scale bar indicates genetic distance. The Pakistani bubaline isolates (1, 2, and 3) clustered closely with Indian and Thai isolates, showing their phylogenetic relationships. The scale bar indicates genetic distance.

**Figure 4 pathogens-14-00066-f004:**
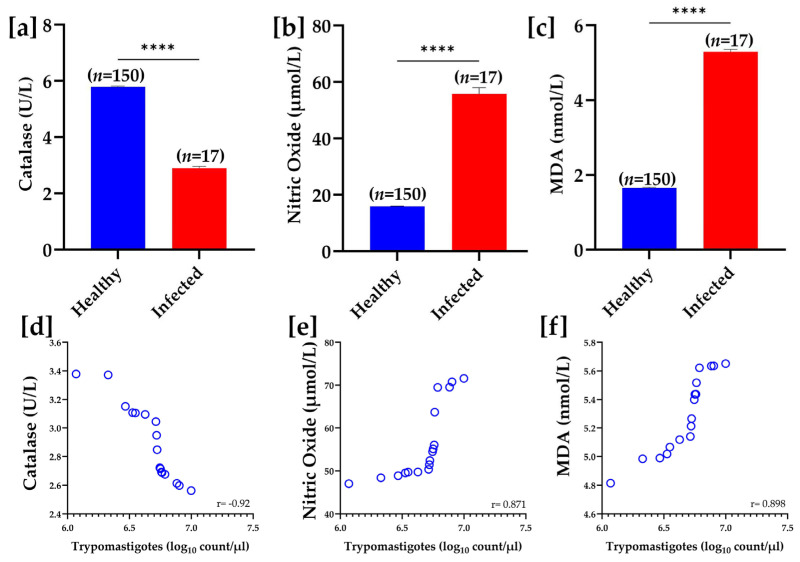
Oxidative stress biomarkers in healthy and infected samples. Panels (**a**–**c**) show bar graphs comparing the levels of catalase (U/L), nitric oxide (µmol/L), and malondialdehyde (nmol/L) between healthy and infected groups. Catalase levels are significantly lower in the infected group (**** *p* < 0.0001), while nitric oxide and MDA levels are significantly higher in the infected group (**** *p* < 0.0001 for both). Panels (**d**–**f**) illustrate the correlation between trypomastigote count (log_10_ count/µL) and levels of catalase, nitric oxide, and MDA, respectively. A strong negative correlation was observed between trypomastigote count and catalase levels (r = −0.92), while positive correlations were recorded regarding nitric oxide (r = 0.871) and MDA levels (r = 0.898).

**Figure 5 pathogens-14-00066-f005:**
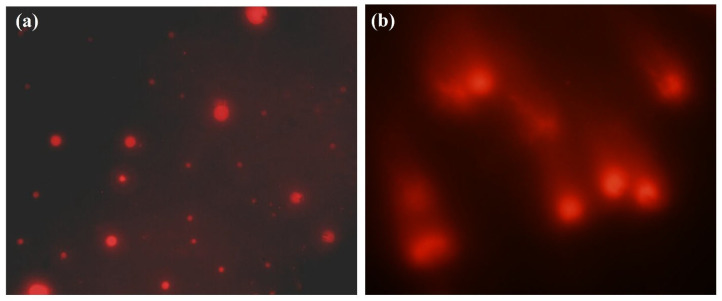
Comet assay results show DNA integrity in buffaloes’ peripheral blood cells. (**a**) Cells from healthy animals exhibited intact, undamaged nuclei and appeared as distinct fluorescent spots. (**b**) DNA damage in cells from *T. evansi*-infected buffaloes exhibiting comet-like appearance with tails indicating DNA fragmentation.

**Figure 6 pathogens-14-00066-f006:**
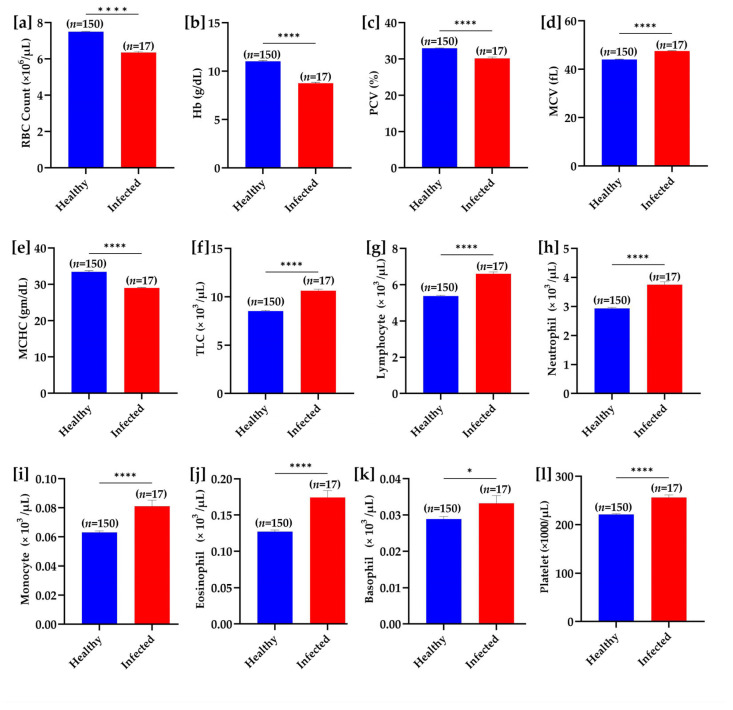
Hematological parameter comparison between healthy and *T. evansi*-infected buffaloes under natural field conditions. (**a**) RBC count, (**b**) hemoglobin (Hb), (**c**) packed cell volume (PCV), (**d**) mean corpuscular volume (MCV), (**e**) mean corpuscular hemoglobin concentration (MCHC), (**f**) total leukocyte count (TLC), (**g**) lymphocyte count, (**h**) neutrophil count, (**i**) monocyte count, (**j**) eosinophil count, (**k**) basophil count, and (**l**) platelet count. Significant differences between groups are marked with asterisks, where * *p* < 0.05 and **** *p* < 0.0001.

**Figure 7 pathogens-14-00066-f007:**
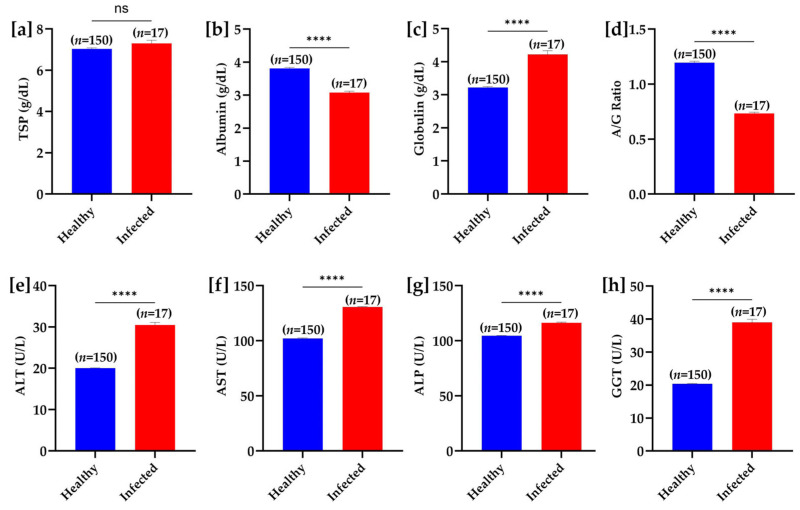
Biochemical parameter comparison between healthy and *T. evansi*-infected buffaloes under natural field conditions. (**a**) Total serum protein (TSP), (**b**) albumin, (**c**) globulin, (**d**) albumin/globulin (A/G) ratio, (**e**) alanine aminotransferase (ALT), (**f**) aspartate aminotransferase (AST), (**g**) alkaline phosphatase (ALP), and (**h**) gamma-glutamyl transferase (GGT). Significant differences between groups are marked with asterisks, where **** *p* < 0.0001; “ns” indicates a non-significant difference.

**Figure 8 pathogens-14-00066-f008:**
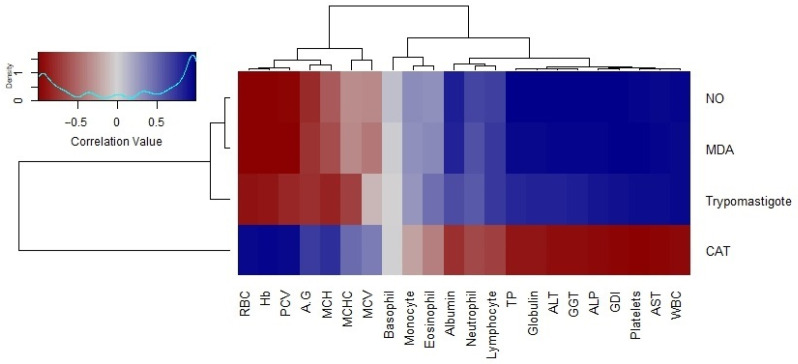
Correlation matrix of oxidative stress markers, trypomastigote counts, and hematobiochemical parameters. The color scale ranges from blue (negative correlation) to red (positive correlation), with the density plot (top-left) indicating the distribution of correlation values. The dendrograms on the left and top represent hierarchical clustering based on patterns of correlation of the parasitemia load, oxidative stress profile, and hematobiochemical parameters, respectively.

**Table 1 pathogens-14-00066-t001:** Detail of primers used in molecular identification of *T. evansi* in infected buffaloes.

Primers	Sequence	Amplicon Size	Reference
ITS-1	F: 5′-CCGGAAGTTCACCGATATTG-3′	480 bp	[18]
	R:5′-TGCTGCGTTCTTCAACGAA-3′		
RoTat 1.2	F: 5′-GCGGGGTGTTTAAAGCAATA-3′	205 bp	[19]
	R: 5′-ATTAGTGCTGCGTGTGTTCG-3′		

**Table 2 pathogens-14-00066-t002:** The incidence rates of *T. evansi* in buffaloes based on field-stained thin (FST) blood film microscopy and PCR using ITS-1 and RoTat 1.2 primers sets. Categories marked by an asterisk (*) were used to calculate the adjusted odds ratios (aOR) and their 95% confidence intervals (CI) for demographic categories. *p*-value describes the significance of the association between risk factors and demographic variables regarding the presence of infection (based on ITS-1 PCR).

	FST	ITS-1	RoTat 1.2	aOR (95%CI)	*p*-Value
	**Gender**				
Female	10 (8.62%)	12 (10.34%)	12 (10.34%)	*	0.91
Male	5 (9.8%)	5 (9.8%)	5 (9.8%)	0.94 (0.31–2.82)	
	**Age**				
<1 year	3 (6%)	5 (10%)	5 (10%)	*	0.8
>4 years	6 (12.5%)	6 (12.5%)	6 (12.5%)	1.3 (0.36–4.53)	
1–4 years	6 (8.7%)	6 (8.7%)	6 (8.7%)	0.86 (0.25–2.98)	

**Table 3 pathogens-14-00066-t003:** Comparison of comet assay results between infected and healthy buffalo lymphocytes. Percentages represent the distribution of DNA damage classes (Class 0 to Class 4) observed in infected versus healthy groups. The mean values of the genetic damage index were significantly higher damage in infected buffaloes compared to healthy samples (*p* < 0.0001).

	Class 0	Class 1	Class 2	Class 3	Class 4	GDI	*p*-Value
Infected	58.56%	27.54%	9.7%	4.19%	2.51%	0.7 ± 0.04	<0.0001
Healthy	83.74%	13.37%	2.89%	0%	0%	0.196 ± 0.004	

## Data Availability

All data are available in the manuscript.

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
