# Peer review of "Molecular Characterization, Oxidative Stress-Mediated Genotoxicity, and Hemato-Biochemical Changes in Domestic Water Buffaloes Naturally Infected with Trypanosoma evansi Under Field Conditions"

_pathogens, 2025, doi:10.3390/pathogens14010066_

Round 1
Reviewer 1 Report
Comments and Suggestions for Authors
the study aims to assess genetic relationship of naturally infecting trypanosomes from specific district in Pakistan and evaluates both parasitesemia and oxidative stress impact on infected animals. The manuscript is clear and well- structured. the cited references are updated. The experimental design is appropriate to test the study hypothesis. The results are resproducible based on the details given in the method section. The tables and figures are mostly clear and easy to understand. The conclusions are consistent with evidence and arguments presented. I have some comments:
- In the results section:
1- It would be easier for audience to put the result in headlines formate such as 3.1, etc..
2- in table 1: please list what FST stands for. Also what does the P value indicate here? I guess the difference between the field stained test and PCR.
page 6, line 189: it is table 1 not 2
3- In figure 2, please mention that the displayed data doesn't include all the tested animals other animals's results are unshown.
4- Authors used bar graph for all data but it would be better to use scatter pattern graphs (there is option in GraphPad prism for that) or list the numbers of animals used in each graph.
5- Figure 5b is not clear and seems not the same scale of 5A.
6- page 9, line 244, it is table 2 not 3.
Author Response
Thank you for your thorough review and insightful comments on our manuscript. We have addressed all the points raised and incorporated the suggested changes to improve the clarity and quality of the work. Below are our detailed responses to each comment.
Comment 1: It would be easier for audience to put the result in headlines format such as 3.1, etc.
Response 1: Thank you for the suggestion. We have revised the results section to present findings in a headline format (e.g., 3.1, 3.2, etc.) for improved clarity and ease of understanding.
Comment 2: in table 1: please list what FST stands for. Also what does the P value indicate here? I guess the difference between the field-stained test and PCR.
Response 2: Thank you for pointing this out. We have added the full form of FST at line No. 128, and in the caption of Table at line 197. Additionally, we clarified that the P-value indicates the significance of the association between risk factors and demographic variables concerning the presence of infection, as determined by PCR (Line 200 and 201).
Comment 3: page 6, line 189: it is table 1 not 2
Response 3: Thank you for highlighting this error. The table reference on page 6, line 189 was actually Table 2. The Table 1 was mistakenly missing during the formatting of manuscript.
The missing Table 1 has now been added (at page 5 b/w line 136 and 137), ensuring the proper sequence and alignment of all tables in the manuscript.
Comment 4: In figure 2, please mention that the displayed data doesn't include all the tested animals other animals's results are unshown.
Response 4: Thank you for your suggestion. We have clarified in the manuscript at line 206 that the data displayed in Figure 2 (a) does not include all the tested animals and results for other animals are not shown.
Comment 5: Authors used bar graph for all data but it would be better to use scatter pattern graphs (there is option in GraphPad prism for that) or list the numbers of animals used in each graph.
Response 5: Thank you for your valuable feedback. While we acknowledge the utility of scatter plots, we chose bar graphs for this analysis as they effectively present the mean ± standard error of the mean (SEM), which is crucial for highlighting central tendencies and variability across the groups. Additionally, bar graphs allow us to clearly indicate statistical significance between groups. For clarity, the number of animals included in each group has been incorporated in the figures.
Comment 6: Figure 5b is not clear and seems not the same scale of 5A.
Response 6: Thank you for your observation. We would like to clarify that the observed appearance of Figure 5b is due to digital magnification, emphasizing the genetic damage captured during the comet assay. We assure you that the image quality is clear and serves to illustrate the described findings effectively. Since the image is an accurate representation of the assay results, we kindly request retaining it in its current form.
Comment 7: page 9, line 244, it is table 2 not 3.
Response 7: Thank you for noting this. The reference on page 9, line 244 has been verified as Table 3. The earlier confusion arose due to the previously missing Table 1, which has now been added, ensuring proper table numbering throughout the manuscript.
Reviewer 2 Report
Comments and Suggestions for Authors
attached

Author Response
Thank you for your time and comments. We have carefully considered your observations and appreciate the opportunity to clarify and address the points. Below are the detailed responses to each of the comments.
Comment 1: The small sample size and lack of clarity in statistical analysis weaken the reliability and generalizability of the findings.
Response 1: Thank you for the observation. While we acknowledge the concern regarding sample size, we slightly disagree with the reviewer. The sample size was determined using a valid sample size calculation formula (Line # 114 to 119) and a randomized multi-stage cluster sampling approach based on the available resources and the preliminary survey in the studied population. We conducted thorough statistical analyses using appropriate tools to ensure the reliability of our findings. Details of the statistical methods, including tests and tools used, are clearly provided in the Materials and Methods section under the heading statistical analysis (Line # 174 to 189).
Comment 2: Correlations between oxidative stress markers, hematobiochemical parameters, and trypomastigote counts are presented without establishing causation or exploring underlying mechanisms. Additionally, the limited phylogenetic analysis does not capture the genetic diversity of T. evansi, and potential influencing factors such as seasonal variations, co-infections, and management practices are not considered.
Response 2: We appreciate the reviewer’s observation and would like to provide further clarification. The housing and management practices in the study area were found to be largely uniform that minimizes the potential variability arising from these factors. Healthy animals were compared in the study to establish a baseline which allowed us to confidently attribute the observed differences in oxidative stress markers, hematobiochemical parameters, correlated with trypomastigote counts of T. evansi infection. The oxidative stress markers (MDA, NO, and catalase), hematological parameters, and biochemical changes were systematically analyzed and compared between infected and healthy groups using appropriate statistical tests. Furthermore, the strong correlations observed between oxidative stress markers and trypomastigote counts provide additional support for concluding the specific link between T. evansi infection and the physiological and biochemical changes which were evident in infected animals compared with the normal healthy ones.
Regarding the concern on phylogenetic analysis, the study was first one to confirm the describe the genetic identity of T. evansi in the study area using ITS-1 primers. The phylogenetic analysis revealed a high genetic similarity with Indian isolates that are consistent with the geographical proximity patterns. Although we recognize that a more comprehensive genetic diversity study incorporating a larger number of isolates and additional molecular markers could provide deeper insights, however, such an endeavor was beyond the scope and resources of the present study. Our analysis has provided a foundational understanding of the genetic characteristics of T. evansi in the region and can guide future studies focusing on population genetics, strain variation, disease epidemiology and control strategies.
Although we acknowledge that seasonal variations and other external factors may have some influence, the uniformity of managemental practices and the inclusion of controls may confound these concerns and further support the reliability of our findings. Regarding co-infections, the thin blood smear microscopy was performed for all samples, and careful examination were performed to rule out other potential blood-borne parasitic infections (mentioned in the Materials and Methods section Line# 125 to 129 of revised manuscript).
Comment 3: The regression analysis lacks detailed explanation of model assumptions, and cross-validation of findings is missing, raising concerns about the robustness and applicability of the results.
Response: We appreciate the comment regarding the regression analysis. We would like to clarify that stepwise backward elimination was used to select the most significant independent variables. Additionally, k-fold cross-validation was performed to validate the model's stability and generalizability (Line # 174 to 189).
Reviewer 3 Report
Comments and Suggestions for Authors
The manuscript intends to elucidate the biochemical alterations and oxidative stress-induced genotoxicities in buffaloes.
The manuscript is adequately detailed but requires modifications.
Here are my observations:
Line 31, 215 - Kindly convert the parasite count to a metric unit rather than a logarithmic scale.
In the inset map, I recommend assigning numbers to the depicted regions and including them in the legend. The names of the locations are confusing.
Line 110 - What is a tehsil?
Line 131 - It is essential to reference the bibliographic sources for the primers ITS-1 and RoTat 1.2. Incorporate these references into the article's bibliography.
Table 1-1. What is the aggregate count of males and females for each age group?
2. Incorporate the acronym FST into the caption. Incorporate the acronym FST into the caption.
Figure 2 - Why is there an absence of a control sample for each gel and primer pair used? What is lane 6 of (b)?
Figure 3 - Should the text emphasise the locations from which the samples sequenced in this study were obtained? Potentially analysing this information.
Line 244 - The correct reference is table 2, not table 3.
Figure 9 - I recommend deleting this figure, as it fails to elucidate any information, or alternatively, presenting the information differently.
Delete the complete citation from the text in lines 312 and 322.
Author Response
We are highly thankful for your valuable feedback and suggestions for enhancing our manuscript. We have carefully considered all the comments and made the necessary revisions accordingly. Please find our point-by-point responses below.
Comment 1: Line 31, 215 - Kindly convert the parasite count to a metric unit rather than a logarithmic scale.
Response 1: Thank you for your suggestion. We have converted the parasite count to a metric unit instead of using the logarithmic scale, as suggested.
Comment 2: In the inset map, I recommend assigning numbers to the depicted regions and including them in the legend. The names of the locations are confusing.
Response 2: Thank you for the suggestion. We have updated the inset map to include the names of the regions directly on the map for improved clarity and to avoid any confusion.
Comment 3: Line 110 - What is a tehsil?
Response 3: Thank you for pointing this out. A tehsil is an administrative division in some South Asian countries, including Pakistan. We have clarified this term in the manuscript (now at Line #113) for better understanding.
Comment 4: Line 131 - It is essential to reference the bibliographic sources for the primers ITS-1 and RoTat 1.2. Incorporate these references into the article's bibliography.
Response 4: Thank you for your observation. The bibliographic sources for the primers ITS-1 and RoTat 1.2 have been incorporated into Table 1 (which was missing previously) as references 18 and 19, and these have also been included in the article's bibliography.
Comment 5: Table 1-1. What is the aggregate count of males and females for each age group? 2. Incorporate the acronym FST into the caption. Incorporate the acronym FST into the caption.
Response 5: Thank you for pointing this out. (1) We have updated the aggregate count in text of manuscript at line # 192 to 195. (2) We have also added the full form of FST at line No. 128, and in the caption of Table at line 197.
Comment 6: Figure 2 - Why is there an absence of a control sample for each gel and primer pair used? What is lane 6 of (b)? Added in caption.
Response 6: Thank you for your observation. Due to limited resources and the fact that the PCR protocols had already been optimized in our laboratory, we did not include a separate control sample for each gel and primer pair. Furthermore, the results were confirmed through sequencing, which served as an additional validation step. As for lane 6 of panel (b), this has been clarified in the updated caption.
Comment 7: Figure 3 - Should the text emphasise the locations from which the samples sequenced in this study were obtained? Potentially analysing this information.
Response 7: We appreciate the reviewer’s insightful comment. The figure caption has been revised to emphasize the Pakistani isolates sequenced in this study, as highlighted in the phylogenetic tree (Line# 222, 225-227). This provides clarity regarding their origin and relationship with isolates from other geographical locations.
Comment 8: Line 244 - The correct reference is table 2, not table 3.
Response 8: Thank you for noting this. The table has been verified as Table 3. The earlier confusion arose due to the previously missing Table 1, which has now been added, ensuring proper table numbering throughout the manuscript.
Comment 9: Figure 9 - I recommend deleting this figure, as it fails to elucidate any information, or alternatively, presenting the information differently. Delete the complete citation from the text in lines 312 and 322.
Response 9: Thank you for your feedback. We have deleted Figure 9, as well as the complete citation from the (now at) lines # 327, 328 and 336, in accordance with your suggestion.
Round 2
Reviewer 3 Report
Comments and Suggestions for Authors
The changes made to the manuscript correspond with the recommendations provided. I advocate for acceptance.